# Exploring Anatomo-Morphometric Characteristics of Infrapatellar, Suprapatellar Fat Pad, and Knee Ligaments in Osteoarthritis Compared to Post-Traumatic Lesions

**DOI:** 10.3390/biomedicines10061369

**Published:** 2022-06-09

**Authors:** Chiara Giulia Fontanella, Elisa Belluzzi, Assunta Pozzuoli, Manuela Scioni, Eleonora Olivotto, Davide Reale, Pietro Ruggieri, Raffaele De Caro, Roberta Ramonda, Emanuele Luigi Carniel, Marta Favero, Veronica Macchi

**Affiliations:** 1Department of Industrial Engineering, University of Padova, 35131 Padova, Italy; chiaragiulia.fontanella@unipd.it; 2Centre for Mechanics of Biological Materials, University of Padova, 35131 Padova, Italy; elisa.belluzzi@unipd.it (E.B.); assunta.pozzuoli@unipd.it (A.P.); pietro.ruggieri@unipd.it (P.R.); raffaele.decaro@unipd.it (R.D.C.); veronica.macchi@unipd.it (V.M.); 3Musculoskeletal Pathology and Oncology Laboratory, Department of Surgery, Oncology and Gastroenterology (DiSCOG), University of Padova, 35128 Padova, Italy; 4Orthopedics and Orthopedic Oncology, Department of Surgery, Oncology and Gastroenterology (DiSCOG), University of Padova, 35128 Padova, Italy; 5Department of Statistical Sciences, University of Padova, 35121 Padova, Italy; manuela.scioni@unipd.it; 6RAMSES Laboratory, RIT Department, IRCCS Istituto Ortopedico Rizzoli, 40136 Bologna, Italy; eleonora.olivotto@ior.it; 7Applied and Translational Research Center (ATRc), IRCCS Istituto Ortopedico Rizzoli, 40136 Bologna, Italy; davide.reale@ior.it; 8Institute of Human Anatomy, Department of Neurosciences, University of Padova, 35121 Padova, Italy; 9Rheumatology Unit, Department of Medicine-DIMED, University—Hospital of Padova, Via Giustiniani, 2, 35128 Padova, Italy; roberta.ramonda@unipd.it (R.R.); faveromarta@gmail.com (M.F.); 10Internal Medicine I, Cà Foncello Hospital, 31100 Treviso, Italy

**Keywords:** osteoarthritis, knee, anterior cruciate ligament rupture, infrapatellar fat pad, suprapatellar fat pad, meniscal tear, magnetic resonance, posterior cruciate ligament, patellar ligament, segmentation

## Abstract

Several studies have investigated cartilage degeneration and inflammatory subchondral bone and synovial membrane changes using magnetic resonance (MR) in osteoarthritis (OA) patients. Conversely, there is a paucity of data exploring the role of knee ligaments, infrapatellar fat pad (IFP), and suprapatellar fat pad (SFP) in knee OA compared to post-traumatic cohorts of patients. Therefore, the aim of this study was to analyze the volumetric and morphometric characteristics of the following joint tissues: IFP (volume, surface, depth, femoral and tibial arch lengths), SFP (volume, surface, oblique, antero–posterior, and cranio–caudal lengths), anterior (ACL) and posterior cruciate ligament (PCL) (volume, surface, and length), and patellar ligament (PL) (volume, surface, arc, depth, and length). Eighty-nine MR images were collected in the following three groups: (a) 32 patients with meniscal tears, (b) 29 patients with ACL rupture (ACLR), and (c) 28 patients affected by end-stage OA. Volume, surface, and length of both ACL and PCL were determined in groups a and c. A statistical decrease of IFP volume, surface, depth, femoral and tibial arch lengths was found in end-stage OA compared to patients with meniscal tear (*p* = 0.002, *p* = 0.008, *p* < 0.0001, *p* = 0.028 and *p* < 0.001, respectively) and patients with ACLR (*p* < 0.0001, *p* < 0.0001, *p* = 0.008 and *p* = 0.011, respectively). An increment of volume and surface SFP was observed in group b compared to both groups a and c, while no differences were found in oblique, antero–posterior, and cranio–caudal lengths of SFP among the groups. No statistical differences were highlighted comparing volume, surface, arc, and length of PL between the groups, while PL depth was observed to be decreased in end-OA patients compared with meniscal tear patients (*p* = 0.023). No statistical differences were observed comparing ACL and PCL lengths between patients undergoing meniscectomy and TKR. Our study confirms that IFP MR morphometric characteristics are different between controls and OA, supporting an important role of IFP in OA pathology and progression in accordance with previously published studies. In addition, PL depth changes seem to be associated with OA pathology. Multivariate analysis confirmed that OA patients had a smaller IFP compared to patients with meniscal tears, confirming its involvement in OA.

## 1. Introduction

Osteoarthritis (OA) is the most common musculoskeletal disorder with an increasing impact worldwide. It is estimated that OA affects more than 32.5 million adults in the US and over 350 million adults globally, leading to both pain and disability [1,2]. OA can damage any joint, but preferentially affects knees, hands, hips, and the spine [2]. The pathogenesis is still under investigation. However, it is well established that OA affects not only cartilage but also all the other joint tissues [3,4,5]. Recently, attention has been focused on the role of the infrapatellar fat pad (IFP) in OA disease, hypothesizing that IFP and synovial membrane might act as an anatomo-functional unit [6]. It has been shown that IFP is a source of adipocytokines, which could contribute to OA inflammation. In addition, aging has a role in the remodeling of this tissue, and OA IFP appeared to be a stiffer adipose tissue compared to subcutaneous, visceral adipose tissues and heel fat pads [7,8,9,10]. Previous research has established that OA IFP is more inflamed, fibrotic, and vascularized compared to IFP isolated from cadavers and patients with anterior ligament cruciate rupture (ACLR) [11,12]. Moreover, OA IFP adipocytes are larger and numerically lower than anterior cruciate ligament (ACL) IFP adipocytes and there is a different collagen-type distribution, which could explain the changes in the biomechanical characteristics found in OA IFP [12,13]. Interestingly, Eymard et al. reported that the OA suprapatellar fat pad (SFP), another knee fat pad, was more fibrotic and produced higher levels of cytokines than subcutaneous adipose tissue in OA patients [8].

While there are numerous studies investigating cartilage, subchondral bone, and synovial membrane changes in OA patients using magnetic resonance imaging (MR) [14,15], only a few papers have been published so far using this imaging technique to explore the role of both IFP and SFP in knee OA. Moreover, there is a lack of studies evaluating volumetric and morphometric characteristics of ACL, posterior cruciate ligament (PCL), and patellar ligament (PL), also known as patellar tendon, in OA patients. 

In 2015, Cowan et al. observed that IFP volume was higher in patients with patellofemoral joint OA compared to controls in a small cohort [16]. On the contrary, no differences were found in IFP volume between OA patients and healthy patients in two studies [17,18]. Cai et al. reported that a greater IFP volume was associated with greater knee cartilage volume and fewer structural abnormalities, suggesting a protective role of IFP size in OA [19]. Pan et al. measured the IFP size at the baseline and after 2.6 years of follow-up in a healthy cohort showing that the IFP size area appeared to have a protective role in the onset of knee symptoms and cartilage damage in older female adults [20]. Teichtahl et al. showed as a larger IFP at baseline was associated with reduced knee pain at follow-up in OA patients [21]. Moreover, a decrease in IFP volume, depth, and femoral and tibial arch lengths was found in moderate and end-stage OA compared to controls [22]. On the contrary, another study did not finc significant differences in IFP morphology between painful versus painless knees of people with unilateral knee pain [23]. 

Other than IFP size, changes in the intensity of the MR signal of the IFP were investigated. Han et al. found an association between the hypointense signal and increased knee cartilage defects, bone marrow lesions as well as knee symptoms in a cross-sectional study following the participants for at least 2 years [24]. A difference in IFP hypointense signal was also found between groups with moderate and end-stage OA compared to controls [22], suggesting a role of fibrosis in OA progression. In addition, Ruhdorfer et al. showed a difference in IFP heterogeneity between OA patients and controls [17]. In 2020, De Vries et al. observed that hyperintense IFP regions show different perfusion in controls, patients with patellofemoral pain, and OA patients, supporting the inflammatory pathogenesis of OA [25]. 

Regarding MR findings on SFP, the studies published so far on OA have been focused on the association between the SFP mass effect (defined as a posterior convex border) and the presence of anterior knee pain with contradictory results [26,27]. SFP signal intensity alterations were associated with OA, pain, and bone marrow lesions [27]. Data from the Osteoarthritis Initiative reported an association between SFP signal intensity alterations and the progression of patellofemoral OA [28]. While a recent small study showed no differences in SFP volumetric and morphometric characteristics between patients with moderate and end-stage OA compared to non-osteoarthritic controls [22]. 

The aim of this study is to investigate and compare the IFP, SFP, ACL, PCL, and PL MR volumetric and morphometric characteristics in patients with meniscal tears, ACLR, and end-stage OA. 

## 2. Materials and Methods

### 2.1. Study Population

Patients with meniscal tears and patients with end-stage OA were enrolled at the Orthopaedic Clinic (University-Hospital of Padova, Padova, Italy), while patients with ACLR were enrolled at the IRCSS Rizzoli Orthopedic Institute (Bologna, Italy). The patients with meniscal tear were obtained in the framework of a multicenter prospective cohort study, entitled “The role of the meniscus in OA pathology and symptoms”, funded by the Italian Ministry of Health between 2012 and 2016 (Project code: GR-2010-2317593) [29]. The study was approved by both the Local Ethical Committee of the University-Hospital of Padova (protocol code 0005073 and CESC code 4510/AO/18) and IRCS Rizzoli of Bologna (prot N.0007206) and patients were enrolled after providing written informed consent. Patients with previous knee surgery or other significant pathologies (such as cancer and other rheumatologic disorders) were excluded from the study. For each patient, demographic and clinical data were retrieved.

The patients were divided into three groups: (a) patients with meniscal tears; (b) patients with ACLR; (c) patients undergoing total knee replacements for end-stage knee OA.

### 2.2. MR Image Acquisition and Analysis

MR imaging of each patient was collected prior to surgery. Images were obtained by different magnetic resonance equipment, and different imaging sequences, but all had at least T1- and T2-weighted sequences, a sequence with fat suppression, and a sequence for each scanning plane (sagittal, coronal, and axial).

#### Volumetric Analysis 

For volumetric analysis, the T1-weighted sequence on the sagittal plane was chosen, with the slice thickness of 4 mm, the matrix of 256 × 256 pixel and the number of slices for each participant to cover the whole knee. Segmentation of the soft tissues were derived manually using an imaging density segmentation software (ITK-SNAP). The reconstruction of the soft tissues requires the elaboration of MR images that makes it possible to create a mask of the tissues and distinguish the different soft tissues based on voxel intensity (Figure 1). Manual mask-based adaptations were applied where necessary. The software automatically integrates the segmented structures in a single 3D image, to calculate the volumes and the surfaces. The lengths were calculated considering a single slice and defining splines construct along highlighted tissues. The same reader (C.G.F.) defined all segmentations, with quality control performed by an experienced radiologist (V.M.) to ensure reliable measurements [30]. 

The IFP volume and surface, the depth, the femoral arch length, and the tibial arch length of the sagittal slice located in the center of the segmented IFP (central sagittal slice) were determined (Figure 2a). The depth was calculated as the length of the perpendicular segment to the patellar tendon passing through the point of the IFP more internally to the joint. The femoral and tibial arch lengths were calculated as the profile of the IFP adjacent to the corresponding bone extremities [22].

In the same way, the SFP volume and surface were determined together with the antero–posterior (A–P), cranio–caudal (C–C), and oblique lengths (OBL) of the sagittal slice located in the center of the segmented IFP (central sagittal slice) (Figure 2b). The A–P length was defined as the distance between the posterior point of the pad and the dorsal contour of the distal quadriceps’ tendon along a line parallel to the patient’s axial plane. The C–C length was defined as the distance between the most superior and inferior points of the pad along a line vertical to the patient’s axial plane. The OBL length was defined as the distance between the most posterior and anterior points of the pad along an obliquely oriented measurement tangent running parallel to the superior aspect of the base of the osseous contour of the patella. The SFP arch length was evaluated on the transversal slice in correspondence with the A–P measure.

The hypointense signal within the IFP was graded on T2-weighted MR images by counting imaging slices with this abnormality as follows: grade 0 = none; grade 1 = 1–2 slices, grade 2 = 3–5 slices, grade 3 = ≥6 slices, as previously reported [22,24]. The hypointense scoring was also assessed within the SFP even if this MR grading system was not validated at this site. Moreover, the presence of edema within the SFP was evaluated, i.e., the diffuse hyperintense signal within the fat pad. These evaluations were conducted by an experienced radiologist (VM).

The ACL and PCL volumes, surfaces, and lengths were extracted (Figure 2c,d) in the meniscal tear group and in the end-stage OA group. ACL could not be measured in the ACLR group as the ligament was ruptured. The ACL and PCL lengths were defined as the maximum length, calculated on the sagittal slice where both the attachment sites of each ligament were visible. The measurements were taken from the midpoint of the ACL and PCL areas. 

The PL volume and surface were determined, while the PL depth and length were evaluated on the sagittal slice located in the center of the segmented PL (central sagittal slice) (Figure 2e). The PL length was calculated as the length of the ligament from the lower pole of the patella to its inferior insertion on the tibia, while the PL depth was calculated as the length of the perpendicular segment to the PL passing through the midpoint of the PL length. The PL arch length was evaluated on the transversal slice in correspondence with the femoral condyles (Figure 2f).

### 2.3. Statistical Analysis

The data were reported as the median and interquartile range (IQR). Chi-square (χ^2^) test or Fisher’s exact test were performed to compare categorical and dichotomous data. The distribution of continuous variables was checked with the Shapiro–Wilk test. One-way ANOVA or Kruskal–Wallis, with Tukey’s post hoc tests, were used to compare continuous variables depending on the distribution of the data. Spearman’s correlations were performed to analyze correlations between variables. Linear regression models were applied to investigate the association between the quantitative variables and some predictors. The Shapiro–Wilk test was used to determine whether residuals were distributed normally. In case of non-normal residuals appropriate transformations of the response variable were modeled. Cumulative link models were used to study the association between ordinal variables and some predictors.

A *p* < 0.05 was considered significant. All analyses were performed with SPSS version 25.0 or R [31].

## 3. Results

### 3.1. Patient Characteristics 

A total of 89 patients were enrolled in this study and divided into three groups. Thirty-two patients had meniscal tears, 29 ACLR, and 28 end-stage OA. Demographic data of the patients are reported in Table 1. A difference was observed regarding gender between the three groups (*p* < 0.001). Patients with meniscal tear and patients with ACLR were younger compared to end-stage OA (*p* < 0.001). Patients with ACLR were younger in comparison to patients with meniscal tears (*p* = 0.016). A difference between patients with ACLR and end-stage OA was observed regarding body mass index (BMI) (*p* < 0.001). IFP, SFP, and ligaments morphometric characteristics were investigated and compared among the three groups.

### 3.2. IFP and SFP Morphometric Characteristics

IFP volume, surface, depth, and femoral and tibial arch lengths were measured and compared between the three groups of patients (Figure 3, Table 2). IFP volume, surface, depth, femoral and tibial arch lengths were higher in patients with meniscal tears compared to end-stage OA patients (*p* = 0.002, *p* = 0.008, *p* < 0.0001, *p* = 0.028, and *p* = 0.0005, respectively). IFP volume, surface, depth, femoral and tibial arch lengths were higher also in patients with ACLR compared to end-stage OA patients (*p* < 0.0001, *p* < 0.0001, *p* < 0.008, and *p* = 0.011, respectively). Interestingly, patients with ACLR exhibited higher IFP volume, surface, depth, and femoral and tibial arch lengths compared to patients with meniscal tears, even if no statistical difference was observed. 

The evaluation of SFP volume, surface, OBL, C–C, A–P, and arch lengths in the three groups (Figure 4) showed that SFP volume and surface were higher in ACLR patients compared to end-stage OA patients (*p* < 0.0001 and *p* < 0.0001, respectively). SFP volume and surface were smaller in patients with meniscal tear compared to patients with ACLR (*p* = 0.001 and *p* = 0.001, respectively). No other differences were highlighted.

IFP and SFP hypointense signals were evaluated in the three groups (Table 3). An increase in IFP hypointense signal was observed in end-stage OA patients compared to both ACLR and meniscal tear patients (*p* < 0.0001). The SFP hypointense signal decreased in ACLR patients compared to meniscal tear patients (*p* = 0.002), while the signal increased in end-stage OA compared to ACLR patients (*p* < 0.005). No differences were reported comparing the presence of SFP edema between the three groups (*p* = 0.385).

### 3.3. ACL, PCL, and PL Morphometric Characteristics

ACL volume, surface, and length were evaluated in patients with meniscal tears and end-stage OA. No differences were observed between the two groups (Figure 5 and Appendix A). PCL volume, surface, and length were measured in all groups without finding any differences. 

PL volume, surface, length, depth, and arc were measured in all the groups showing no differences with the only exception being PL depth (Figure 6, Table 4). PL depth was higher in patients with meniscal tear compared to end-stage OA patients (*p* = 0.023).

### 3.4. Correlation Matrices

The relationships between the variables (age, BMI, and IFP, SFP, ACL, PCL, and PL lengths) were studied building a correlation matrix for all patients (Figure 7a) and each subgroup of patients (Figure 7b–d). 

Age negatively correlated with IFP volume, IFP surface, IFP depth, IFP femoral, tibial arch, and PL depth considering all the patients (r = −0.515, *p* < 0.0001; r = −0.492, *p* < 0.0001; r = −0.490, *p* < 0.0001; r = −0.262, *p* = 0.015; r = −0.301, *p* = 0.005; r = −0.363, *p* = 0.001, respectively). However, when the patients were analyzed separately, the correlations with age were not retained. BMI was positively correlated with SFP OBL (r = 0.268, *p* = 0.015). Strong positive correlations were found between IFP volume and PL volume and PL surface (r = 0.674, *p* < 0.0001; r = 0.557, *p* < 0.0001, respectively). 

### 3.5. Age, BMI, and Gender Influence

Since there is a difference between the three groups regarding gender, age, and BMI and all of these are risk factors for OA, linear models were applied to control their influence on IFP and SFP measurements (Table 5). The volume and surface of SFP did not fit a normal distribution and thus, data were log-transformed before being modeled.

Age did not affect the variables, while gender was always significant with the only exception being the SFP hypointense signal. BMI was significant only regarding IFP surface, SFP volume, and surface. 

Considering the possible confounders, end-stage OA patients exhibited significantly smaller IFP volume, IFP surface, IFP depth, IF tibial arch length, and IFP hypointense signal compared to patients with meniscal tears. 

ACLR patients had a significantly higher volume and surface of both IFP and SFP and lower SFP hypointense signal compared to patients with meniscal tears. The full models are reported in the Appendix A.

## 4. Discussion

In this study, IFP, SFP, ACL, PCL, and PL MR volumetric and morphometric characteristics in patients with meniscal tears, patients with ACLR, and patients with end-stage OA were compared. 

In this work, IFP and SFP geometries were measured by drawing tissue contours and the volumes were computed by a software program, adopting the same procedure as other authors [19,22,32]. Cheruvu et al. determined IFP volume using MR images by using a 3d Reconstruction software, ellipsoidal approximation, and a MATLAB code [32]. Their results showed that there was no significant difference between methods, validating all these procedures to measure the IFP volume [32]. We observed a decrease in IFP volume, surface, depth, and femoral and tibial arch lengths in end-stage OA compared to patients with meniscal tears, confirming the data of our previous study [22]. Moreover, a difference was found also comparing all the IFP lengths of ACLR patients with end-stage OA patients supporting the hypothesis that the IFP volumetric and morphometric characteristics undergoing modifications in patients affected by OA. An increase in the IFP hypointense signal was observed in end-stage OA patients compared to meniscal tear and ACLR patients, confirming that fibrotic IFP changes are important features of OA pathogenesis [12]. Regarding SFP, we observed a difference in SFP volume and surface between patients with ACLR and patients with meniscal tears as well as between patients with ACLR and end-stage OA patients. On the contrary, we did not find differences between patients with meniscal tears and end-stage OA patients suggesting that this fat pad is less involved in OA than IFP, in agreement with our previous study, probably due to the different locations in the knee and to different involvement in joint kinetics [22]. In any case, an increase in SFP hypointense signal was recorded comparing meniscal tear with ACLR patients and ACLR with end-stage OA patients, while no differences were highlighted regarding SFP edema. ACL and PCL morphometric characteristics in patients with meniscal tear, ACLR, and end-stage OA were comparable with the only exception of an increase of PL depth in meniscal tear patients compared to end-stage OA patients. 

In our study, we showed that the IFP geometry of ACLR patients tends to be higher compared to those of meniscal tear patients, even if no statistical differences were highlighted. These results are in line with those of Cheruvu et al. reporting that patients with torn ACLs had significantly larger fat pads compared to a group of age, gender, and sport activity matched controls with intact ligaments [32]. Abnormalities of IFP, such as focal and diffuse edema, tears, scars, and synovial proliferation, are reported to be more common in patients with ACLR than in healthy subjects [33]. The main functional role of the ACL is to provide stability against anterior tibial translation and internal rotation. Patients with ACLR exhibit anterior translation of the tibia relative to the femur [34]. It is presumable that the intact ACL maintains confined the IFP in its localization and upon rupture, the adipose tissue is able to remodel itself. Moreover, the mechanical transmission of loads due to ligament injury causes a change of stress distribution in the tissues, with a consequent modification of its morphometry. Further studies are needed to determine the effects of ACLR on IFP.

Importantly, multivariate analysis confirmed a difference in IFP geometry (except for IFP femoral arch length) between patients with meniscal tears and end-stage OA patients considering possible confounders (age, BMI, and gender). A difference in IFP volume and IFP surface was also found between patients with meniscal tears and patients with ACLR. Interestingly, we found that gender influenced all IFP measures. This is in agreement with the findings of Diepold et al. who observed an increase of IFP volume in healthy men compared to healthy women [35]. 

SFP general linear models confirmed a difference only when comparing patients with ACLR and patients with meniscal tears considering possible confounders (age, BMI, and gender). Thus, it can be hypothesized that this fat pad might not be involved in OA pathology as we have previously described [22]. It should be noted that in literature there is a paucity of MR-based studies on SFP. Schwaiger et al. suggested an association between SPF abnormalities and the progression of patellofemoral OA [28]. However, Schwaiger evaluated the SFP signal alteration and not SFP geometry. An SFP mass effect was described in literature as an SFP expansion with a mass effect on the suprapatellar joint recess, defined by the presence of a convex posterior fat-pad border on the sagittal intermediate-weighted images [26]. Tsavalas et al. reported no correlations between the presence of SFP mass effect and patellofemoral malalignment, or patellofemoral joint OA, or pain [26]. On the contrary, Wang et al. observed that SFP mass effect and/or signal intensity alterations are deleteriously associated with knee pain and radiographic OA [27].

SFP abnormalities as well as IFP abnormalities were studied by Heilmeier et al. in a small cohort of patients with ACLR [36]. While they found several correlations between IFP abnormalities and inflammatory cytokines and metalloproteinases in synovial fluid, SFP abnormalities correlated only with interleukin-6 [36]. However, the authors did not evaluate the volume and surface of SFP.

In our study, SFP does not appear to be involved in OA while it may be affected by the mechanical change induced in the knee because of ACLR.

Several studies have been published on stem cells derived from IFP (for a review see [37]). However, it has been suggested that OA IFP-derived stem cells are primed by the pathological environment and thus these cells seem to exert an incomplete protective activity from OA inflammation [38]. Since SFP seems not to be involved in OA, SFP-derived stem cells are a promising possible source for cell therapeutic solutions to joint degeneration. This idea is supported by the study of Ignacio Muñoz-Criado et al. [39]. They showed that adipose stem cells derived from SFP represent a promising source for cartilage regeneration by promoting efficient endogenous chondrogenesis in a mouse model of severe OA [39].

No differences were observed in ACL and PCL measurements between the groups. Taneja et al. studied PCL morphometry and concluded that subjects with ACLR present larger PCL dimensions compared to controls [40]. Similarly, Jamison et al. analyzed left- and right-sided ACL volumes in control subjects [41]. All these studies measured ACL and PCL geometry from coronal and sagittal slices of MR, using a different procedure compared to our study. Indeed, in the present work ACL and PCL were selected considering only sagittal slices of MR obtaining smaller ACL and PCL volumes in all three groups, probably due to the different methodology. Other authors evaluated the double bundle anatomy of the ACL by measurement of consecutive coronal images using 3D analysis software [42,43]. Fayad et al. studied gender differences in ACL and PCL volumes and showed that gender differences in ACL volume are present, but may be accounted for height differences between males and females [42]. The mean ACL length reported by Van Zyl et al. was 40.6 mm in subjects with no apparent knee pathology, surgery, or trauma [44]. In our study, we observed a mean ACL length of 35.2 mm for patients with meniscal tears and 33.9 for end-stage OA patients). This might be due to the degenerative changes that all joint tissues, including the ACL, undergo during the OA process.

Regarding PCL, which is a knee fundamental stabilizer [45], no differences were observed in terms of volume and length. Ranmuthu et al. found significant differences in the distribution of T1rho and T2 values of the cruciate ligaments according to the sub-region between control and OA [46]. Particularly, in OA both T1rho and T2 values were significantly higher in the distal ACL when compared to the rest of the ligament, while the variation of T2 values within the PCL was lower in OA knees [46]. Thus, it is likely that OA determines changes in ligament composition heterogeneously within the ligament as demonstrated by histological analysis [47], but does not alter the volume and lengths. To the best of our knowledge, this study is the first report analyzing PCL volume and lengths. 

Few studies evaluated the PL length reporting a range of 32–61 mm in healthy individuals [48,49] in line with the findings of this study. The PL length of OA patients was coherent with that reported by Lemon et al. who compared the PL length changes between patients who had their IFP either preserved or excised after total knee replacement [50]. Yoo et al. described the geometry of PL measured on knee MR for not OA patients, analyzing the proximal and distal part of PL. The PL depth was calculated on axial images and ranged from 3.2 to 5.0 mm, for the proximal and the distal parts, respectively. On the contrary, we evaluate the PL depth in the midpoint of the PL length on the sagittal image, measuring an intermediate value. Yoo et al. calculated the PL length as the minimum distance on PL between rotula and tibial insertions [49]. The PL longitudinal length was 40.2 mm [49], while in our study the values for control knees were higher, probably due to the different endpoints chosen for the evaluation of the distance. Wang et al. studied the geometric data of healthy PL and ACL by MR and analyzed the correlation of the two with body weight, height, and gender [51]. They found that the lengths of the PL and ACL in males were significantly greater than those in females [51]. Similarly, to our study, the length of PL was calculated in the ACL and non-ACL injury groups by Kang et al., finding not significant differences [52]. The biomechanical equilibrium of the joints is strongly affected by the presence of trauma or pathologies. In patients with meniscal tear, ACLR and end-stage OA, the biomechanics of the knee is modified and this alteration could affect both spaces and forces, causing both IFP and PL to change. This is evidenced by the positive correlations found between IFP and PL volumes in the three groups. To the best of our knowledge, no data about PL surface and volume are reported in the literature.

The main limitations of this study are the small sample size of patients analyzed and the different facilities used for collecting MR. However, it was possible to perform multivariate analysis, which enabled us to evaluate the influence of demographic confounders on IFP, SFP, and ligament measurements, strengthening the results. Finally, another limitation is the lack of MR collected from a control group of healthy subjects with an active lifestyle and without history of injuries/trauma involving the knees.

## 5. Conclusions

In conclusion, a decrease in IFP volume, surface, depth, femoral and tibial arch lengths, and an increase of the IFP hypointense signal was observed in end-stage OA patients compared to meniscal tear and ACLR patients. These data point out that IFP volumetric and morphometric characteristics are modified by OA disease and that fibrotic changes are important features of OA pathology. No differences between patients with meniscal tears or ACLR and end-stage OA patients were found regarding SFP, suggesting that probably this fat pad is less involved in OA than IFP. Regarding ACL, a decrease in PL depth in end-stage OA patients was observed. 

## Figures and Tables

**Figure 1 biomedicines-10-01369-f001:**
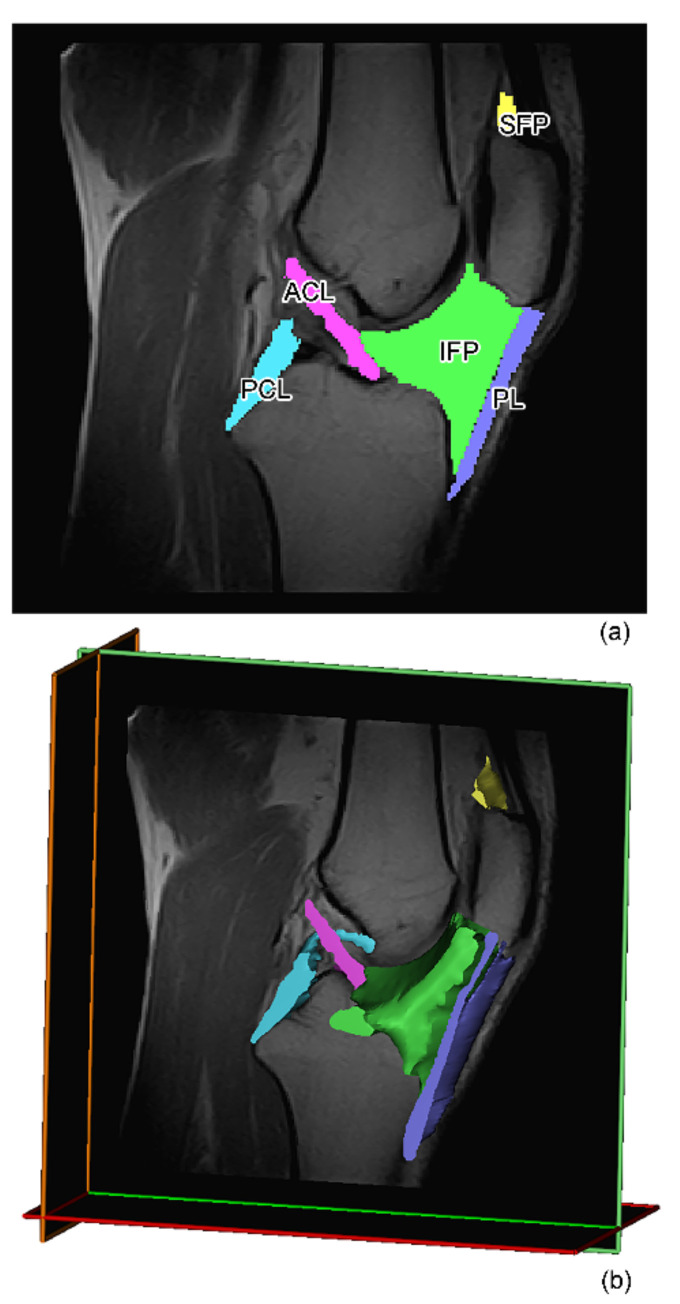
(**a**) IFP (green area), SFP (yellow area), PCL (cyan area), ACL (fuchsia area), and PL (violet area) masks on the central sagittal slice and (**b**) volumes identification on MR of all the tissues analyzed. IFP: infrapatellar fat pad, SFP: suprapatellar fat pad, ACL: anterior cruciate ligament, PCL: posterior cruciate ligament, PL: patellar ligament, MR: magnetic resonance.

**Figure 2 biomedicines-10-01369-f002:**
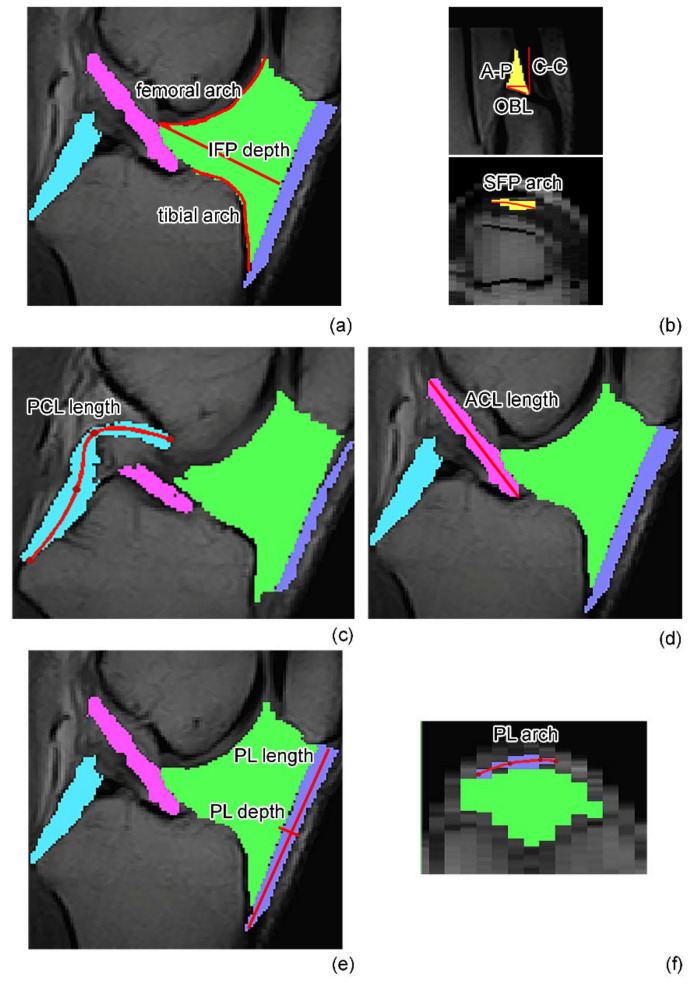
(**a**) Definition of IFP depth, femoral and tibial arch lengths in the central slice. (**b**) Definition of antero–posterior (A–P), cranio–caudal (C–C), and oblique (OBL) lengths for the SFP in the central slice, with the definition of SFP arch length on the transversal slice. (**c**) Definition of PCL and (**d**) ACL lengths on two different sagittal slices. (**e**) Definition of PL depth and length on the central sagittal slice and (**f**) PL arch length on the transversal slice. IFP: infrapatellar fat pad, SFP: suprapatellar fat pad, ACL: anterior cruciate ligament, PCL: posterior cruciate ligament, PL: patellar ligament.

**Figure 3 biomedicines-10-01369-f003:**
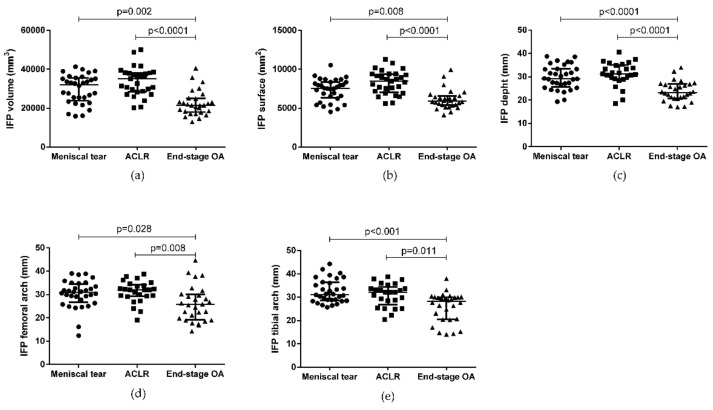
IFP morphometric characteristics in patients with meniscal tear, ACLR, and end-stage OA. (**a**) IFP volume, (**b**) IFP surface, (**c**) IFP depth, (**d**) IFP femoral arch, and (**e**) IFP tibial arch. Data are expressed as median and interquartile range. IFP: infrapatellar fat pad. ACLR: anterior ligament cruciate rupture.

**Figure 4 biomedicines-10-01369-f004:**
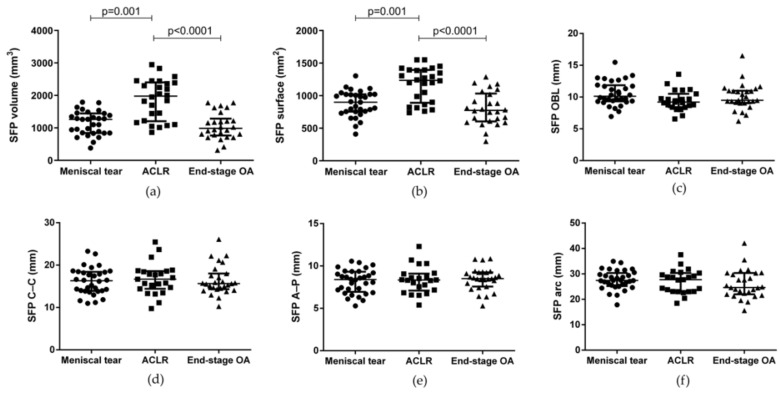
SFP morphometric characteristics in patients with meniscal tear, ACLR, and end-stage OA. (**a**) SFP volume, (**b**) SFP surface, (**c**) SFP OBL, (**d**) SFP C–C (**e**) SFP A–P, and (**f**) SFP arch. Data are expressed as median (interquartile range). SFP: suprapatellar fat pad. ACLR: anterior ligament cruciate rupture. SFP= suprapatellar fat pad. OBL: oblique length. C–C: cranio–caudal length. A–P: antero–posterior length.

**Figure 5 biomedicines-10-01369-f005:**
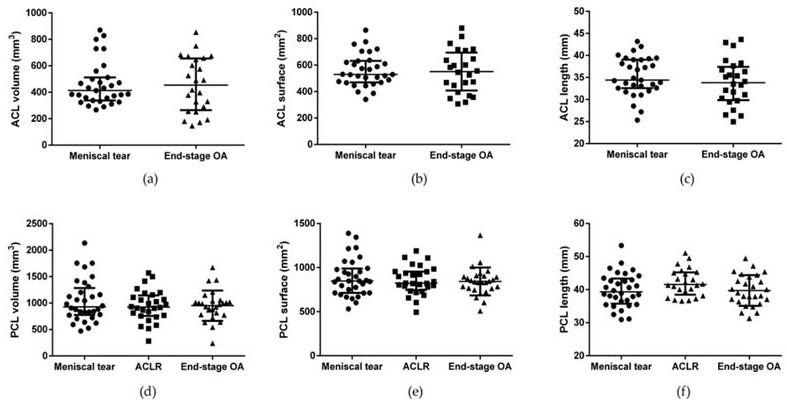
ACL and PCL morphometric characteristics in patients with meniscal tear, ACLR, and end-stage OA. (**a**) ACL volume, (**b**) ACL surface, (**c**) ACL length, (**d**) PCL volume, (**e**) PCL surface, and (**f**) PCL length. Data are expressed as median (interquartile range). ACL: anterior cruciate ligament. PCL: posterior cruciate ligament. ACLR: anterior ligament cruciate rupture.

**Figure 6 biomedicines-10-01369-f006:**
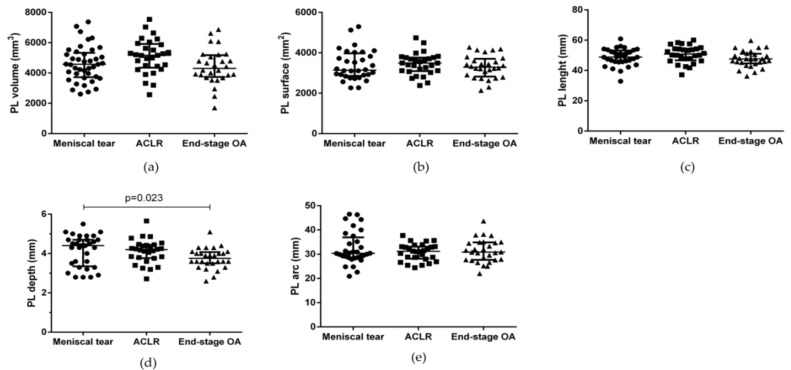
PL morphometric characteristics in patients with meniscal tear, ACLR, and end-stage OA. (**a**) PL volume, (**b**) PL surface, (**c**) PL length, (**d**) PL depth, and (**e**) PL arc. Data are expressed as median (interquartile range). PL: patellar ligament. ACLR: anterior ligament cruciate rupture.

**Figure 7 biomedicines-10-01369-f007:**
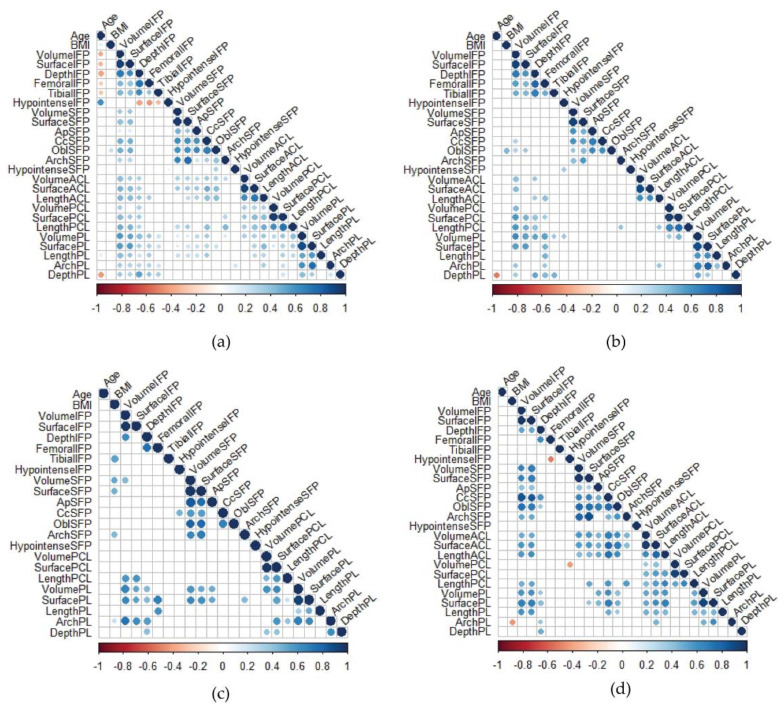
Spearman correlations among the variables. (**a**) all patients, (**b**) meniscal tear patients, (**c**) ACLR patients, and (**d**) end-stage OA patients. Negative correlations are displayed in red color, while positive correlations are in blue. The color intensity and the size of the circle are proportional to the correlation coefficients. ACLR: anterior ligament cruciate rupture.

**Table 1 biomedicines-10-01369-t001:** Demographic data of enrolled patients.

	Meniscal Tear	ACLR	End-Stage OA	*p*-Value
Number of patients	32	29	28	
Sex, male number (%)	20 (62.5)	21 (72.4)	7 (25)	<0.001
Age, years, median (IQR)	48.0 (58.0–35.0)	34.0 (43.5–24.5)	73.5 (78.0–66.5)	^a^ 0.016 ^b^ <0.001 ^c^ <0.001
BMI, Kg/m^2^, median (IQR)	25.1 (29.9–23.8)	25.2 (26.3–22.2)	28.2 (31.6–24.8)	^c^ <0.001

ACLR: anterior cruciate ligament rupture, OA: osteoarthritis, IQR: interquartile range, BMI: body mass index. Data are expressed as median (IQR). ^a^ meniscal tear versus ACLR, ^b^ meniscal tear versus end-stage OA, ^c^ ACLR versus end-stage OA.

**Table 2 biomedicines-10-01369-t002:** IFP and SFP MR morphometric characteristics in patients with meniscal tear, ACLR, and end-stage OA.

	Meniscal Tear	ACLR	End-Stage OA	*p*-Value
IFP volume (mm^3^)	32113 (35460–23943)	35073 (38249–28536)	21552 (25111–18017)	^a^ 0.250 ^b^ 0.002 ^c^ <0.0001
IFP surface (mm^2^)	7550 (8366–6319)	8495 (9393–7062)	5906 (6549–5405)	^a^ 0.074 ^b^ 0.008 ^c^ <0.0001
IFP depth (mm)	29.2 (33.4–25.7)	31.3 (35.1–29.0)	23.2 (26.8–20.8)	^a^ 0.508 ^b^ <0.0001 ^c^ <0.0001
IFP Femoral arch length (mm)	30.9 (34.4–26.7)	32.0 (34.2–29.2)	25.7 (30.1–19.2)	^a^ 0.999 ^b^ 0.028 ^c^ 0.008
IFP Tibial arch length (mm)	31.2 (36.4–28.6)	32.1 (34.4–26.9)	28.3 (30.0–20.5)	^a^ 0.999 ^b^ <0.001 ^c^ 0.011
SFP volume (mm^3^)	1264.2 (1454.0–842.2)	1979.3 (2401.5–1203.9)	1021.6 (1480.0–784.0)	^a^ 0.001 ^b^ 0.999 ^c^ <0.0001
SFP surface (mm^2^)	903.3 (1021.3–754.8)	1236.3 (1395.1–890.1)	782.0 (1066.1–608.8)	^a^ 0.001 ^b^ 0.999 ^c^ <0.0001
SFP OBL (mm)	10.1 (11.8–9.3)	9.2 (10.5–8.4)	9.5 (11.0–9.1)	0.113
SFP C–C (mm)	16.3 (18.4–13.9)	16.7 (18.6–14.4)	15.6 (18.0–14.3)	0.837
SFP A–P (mm)	8.4 (9.3–6.9)	8.3 (9.1–6.8)	8.5 (9.2–7.6)	0.689
SFP arch (mm)	27.4 (30.4–27.4)	27.7 (30.4–23.1)	24.6 (30.4–21.9)	0.380

IFP: infrapatellar fat pad, SFP: suprapatellar fat pad, MR: magnetic resonance, OA: osteoarthritis, OBL: oblique length. C–C: cranio–caudal length. A–P: antero–posterior length. Data are expressed as median (interquartile range). ^a^ meniscal tear vs. ACLR, ^b^ meniscal tear vs. end-stage OA, ^c^ ACLR vs. end-stage OA.

**Table 3 biomedicines-10-01369-t003:** IFP and SFP MR hypointense signal grading in meniscal tear, ACLR, and end-stage OA patients.

	Meniscal Tear (n = 32)	ACLR (n = 29)	End-Stage OA (n = 28)	*p*-Value
IFP				<0.0001
Grade 0	19 (59.4%)	15 (51.7%)	1 (3.6%)	^a^ 0.101
Grade 1	12 (37.5%)	7 (24.1%)	8 (28.6%)	^b^ <0.0001
Grade 2	1 (3.1%)	6 (20.7%)	10 (20.7%)	^c^ <0.0001
Grade 3	0	1 (3.4%)	9 (32.1%)	
	Meniscal tear (n = 32)	ACLR (n = 26)	End-stage OA (n = 27)	*p*-value
SFP				<0.012
Grade 0	8 (25%)	13 (50%)	8 (28.6%)	^a^ 0.002
Grade 1	12 (37.5%)	13 (50%)	10 (37%)	^b^ 0.910
Grade 2	12 (37.5%)	0	9 (32.1%)	^c^ 0.005
Grade 3	0	0	0	
	Meniscal tear (n = 32)	ACLR (n = 24)	End-stage OA (n = 26)	*p*-value
SFP edema				0.385
Yes	12 (37.5%)	5 (20.8%)	9 (34.6%)	
No	20 (62.5%)	19 (79.2%)	17 (65.4%)	

Data are reported as the number of patients (%). ^a^ meniscal tear vs. ACLR, ^b^ meniscal tear vs. end-stage OA, ^c^ ACLR vs. end-stage OA.

**Table 4 biomedicines-10-01369-t004:** PL characteristics in patients with meniscal tear, ACLR, and end-stage OA.

	Meniscal Tear	ACLR	End-Stage OA	*p*-Value
PL volume (mm^3^)	4476 (5242–3659)	5172 (5913–4351)	4302 (5175–3735)	0.076
PL surface (mm^2^)	3139 (3973–2856)	3494 (3764–3093)	3287 (3704–2821)	0.716
PL length (mm)	48.8 (53.1–45.8)	50.8 (54.4–46.7)	47.6 (51.0–44.6)	0.108
PL depth (mm)	4.4 (4.7–3.3)	4.2 (4.4–3.8)	3.7 (4.1–3.5)	^a^ 0.999 ^b^ 0.023 ^c^ 0.153
PL arch (mm)	30.3 (36.9–28.6)	31.2 (33.1–28.2)	30.8 (34.8–27.6)	0.962

PL: patellar ligament, ACLR: anterior ligament cruciate rupture. OA: osteoarthritis. Data are expressed as median (interquartile range). ^a^ meniscal tear vs. ACLR, ^b^ meniscal tear vs. end-stage OA, ^c^ ACLR vs. end-stage OA.

**Table 5 biomedicines-10-01369-t005:** General linear models.

	Age (*p*-Value)	Gender (*p*-Value)	BMI (*p*-Value)	ACLR vs. Meniscal Tear (*p*-Value)	End-Stage OA vs. Meniscal Tear (*p*-Value)
IFP volume	0.579	<0.0001	0.066	0.010	0.018
IFP surface	0.549	<0.0001	0.028	0.002	0.045
IFP depth	0.413	<0.0001	0.907	0.322	0.004
IFP Femoral arch length	0.145	0.0012	0.540	0.457	0.059
IFP Tibial arch length	0.217	<0.001	0.621	0.341	<0.001
SFP volume	0.243	0.025	0.025	<0.001	0.288
SFP surface	0.279	0.018	0.035	<0.001	0.196
IFP hypotense signal	0.662	0.011	0.590	0.225	<0.001
SFP hypotense signal	0.319	0.609	0.832	<0.005	0.598
PL depth	0.082	<0.001	0.229	0.137	0.659

## Data Availability

The data presented in this study are available on request from the corresponding author.

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
