# Peer review of "Exploring Anatomo-Morphometric Characteristics of Infrapatellar, Suprapatellar Fat Pad, and Knee Ligaments in Osteoarthritis Compared to Post-Traumatic Lesions"

_biomedicines, 2022, doi:10.3390/biomedicines10061369_

Round 1

Reviewer 1 Report

The revised version is sufficiently improved

Author Response

We would like to thank the Reviewer for taking the time and effort to consider our manuscript and for the positive comments concerning our research activity.

Reviewer 2 Report

This manuscript is the very interesting research about volumetric and morphometric characteristics of fat pad around the knee. I have checked the revised version. If authors can't use normal knees for control, they need to add that to the limitation.

Author Response

We would like to thank the Reviewer for taking the time and effort to consider our manuscript and for the

positive comments concerning our research activity. We have added in the last paragraph of the manuscript, a limitation concerning the lack of MR from a control group of healthy subjects.

“Finally, another limitation is the lack of MR collected from a control group of healthy subjects with an active lifestyle and without history of injuries/trauma involving the knees.”

This manuscript is a resubmission of an earlier submission. The following is a list of the peer review reports and author responses from that submission.

Round 1

Reviewer 1 Report

Please provide the exact P value in the Abstract, not just a statement that it was <0.05. It is not ideal that two institutions enrolled two different pathologies, as this raises the question of direct comparability. I am slightly puzzled by the pixelization of your images, was there not a better way to delineate the shapes? A function would be much better than the pixels – this is imprecise. Did you perform any sensitivity/repeatability/validity [measuirng the same person several times and comparing the results]? The statistics is slightly odd, mixing ANOVA and KW, invoking regression (which is impossible in KW, non-parametrics), so please provide more explanations or correct them accordingly. Why both SPSS and R? Figure 3 contains P values with more than three decimal digits, please reduce to three. Table 2, never show a P as 1.000, at the very least show ~0.999. Several footnotes state the P<0.05 was considered significant, but this is already said in MM; please remove from these footers. Well, the overall extent of differences between the groups, plus multiple testing (suggesting numerous false positives, or at least increased risk for the false positive results) suggest that you should consider all this a bit more conservative. Try to focus on P values that are well below 0.001, and discuss more such results. The ones that are only slightly below 0.05, you might consider treating as less reliable, and not dedicate much attention to them – only use them in very speculative discussion elements. Figure 7 is severely confounded by the obvious differences among your groups, so I would not be overly keen to show this. Table 4 is again confounded, although you intended to use regression to fix it; of course that the greater tears and more degenerative advancement will have worse indices; you should consider stratification or some other way to show these more clearly. Can you establish their clinical grading or some similar measure, and then compare within such groups? This is just an idea, not a manuscript suggestion. Figure 8 should go to Results. You state that it was possible to create multivariate models, but there is no model diagnostics to be seen, so that statement is false. Please rephrase the Conclusions to something more easily readable, many abbreviations make it difficult to understand the main message.

Reviewer 2 Report

This manuscript is the very interesting research about volumetric and morphometric characteristics of fat pad around the knee. I think the point of view is very good and the image measurement technique is very interesting. However, since the only cases being compared are those that have undergone surgery, the influence of surgical invasion cannot be ruled out, and I think that nothing can be determined based on these results alone. This paper is not suitable for publication in the Journal of Biomedicines.

Overall comments

Since the only cases being compared are those that have undergone surgery, the influence of surgical invasion cannot be ruled out, and I think that nothing can be determined based on these results alone.

Materials and Methods

It is unclear at what point after surgery each imaging was taken. I think it is essential to state the timing of the imaging in order to consider the effects of the surgery. And above all, I think that it is necessary to examine the normal knee as a control.